# Combined Delivery of miR-15/16 through Humanized Ferritin Nanocages for the Treatment of Chronic Lymphocytic Leukemia

**DOI:** 10.3390/pharmaceutics16030402

**Published:** 2024-03-14

**Authors:** Francesca Romana Liberati, Sara Di Russo, Lorenzo Barolo, Giovanna Peruzzi, Maria Vittoria Farina, Sharon Spizzichino, Federica Di Fonzo, Deborah Quaglio, Luca Pisano, Bruno Botta, Alessandra Giorgi, Alberto Boffi, Francesca Cutruzzolà, Alessio Paone, Paola Baiocco

**Affiliations:** 1Department of Biochemical Sciences “Alessandro Rossi Fanelli”, Sapienza University of Rome, P.le A. Moro 5, 00185 Rome, Italy; francescaromana.liberati@uniroma1.it (F.R.L.); sara.dirusso@uniroma1.it (S.D.R.); lorenzo.barolo@uniroma1.it (L.B.); mariavittoria.farina@uniroma1.it (M.V.F.); sharon.spizzichino@uniroma1.it (S.S.); federica.difonzo@uniroma1.it (F.D.F.); alessandra.giorgi@uniroma1.it (A.G.); alberto.boffi@uniroma1.it (A.B.); francesca.cutruzzola@uniroma1.it (F.C.); 2Center for Life Nano- & Neuro-Science@Sapienza, Istituto Italiano di Tecnologia, V.le Regina Elena 291, 00161 Rome, Italy; giovanna.peruzzi@iit.it; 3Department of Chemistry and Technology of Drugs, Sapienza University of Rome, P.le A. Moro 5, 00185 Rome, Italy; deborah.quaglio@uniroma1.it (D.Q.); luca.pisano@uniroma1.it (L.P.); 4University of Rome UnitelmaSapienza, Piazza Sassari 4, 00161 Rome, Italy; bruno.botta@uniroma1.it; 5Laboratory Affiliated to Istituto Pasteur Italia-Fondazione Cenci Bolognetti, Department of Biochemical Sciences “Alessandro Rossi Fanelli”, P.le A. Moro 5, 00185 Rome, Italy

**Keywords:** ferritin nanocages, Bcl-2, miR-15a, miR-16-1 delivery, leukemia, self-assembly

## Abstract

Chronic lymphocytic leukemia (CLL) is a widespread type of leukemia that predominantly targets B lymphocytes, undermining the balance between cell proliferation and apoptosis. In healthy B cells, miR-15/16, a tandem of microRNAs, functions as a tumor suppressor, curbing the expression of the antiapoptotic B cell lymphoma 2 protein (Bcl-2). Conversely, in CLL patients, a recurring deletion on chromosome 13q14, home to the miR15-a and miR16-1 genes, results in Bcl-2 overexpression, thereby fostering the onset of the pathology. In the present research, a novel approach utilizing humanized ferritin-based nanoparticles was employed to successfully deliver miR15-a and miR-16-1 into MEG01 cells, a model characterized by the classic CLL deletion and overexpression of the human ferritin receptor (TfR1). The loaded miR15-a and miR16-1, housed within modified HumAfFt, were efficiently internalized via the MEG01 cells and properly directed into the cytoplasm. Impressively, the concurrent application of miR15-a and miR16-1 demonstrated a robust capacity to induce apoptosis through the reduction in Bcl-2 expression levels. This technology, employing RNA-loaded ferritin nanoparticles, hints at promising directions in the battle against CLL, bridging the substantial gap left by traditional transfection agents and indicating a pathway that may offer hope for more effective treatments.

## 1. Introduction

Numerous studies have highlighted how the dysregulation of gene expression, mediated by miRNA activity, is intricately linked to various facets of cancer biology, underscoring its critical role in the disease progression and behavior, including tumor cell growth, resistance to cell death, and metabolic dysregulation [1,2,3,4,5]. An illustrative example of this phenomenon is the Mir-15/16 family of microRNAs. A significant reduction in the expression levels of miR-15a and miR-16-1 due to the deletion of chromosome 13q14, which commonly occurs in chronic lymphocytic leukemia (CLL) patients, was initially observed by Calin et al. [6,7]. Significant progress has been made in elucidating the involvement of miR-15/16 in regulating the expression of various genes associated with cell proliferation, apoptosis, and the cell cycle in normal B cells, conferring to them a pivotal role of regulators of the anti-apoptotic B-cell lymphoma 2 (Bcl-2) protein [8]. These miRNAs are currently considered biomarkers for assessing the onset and progression of B-CLL and are promising candidates for therapeutic interventions as tumor suppressors [9,10,11,12]. The loss of miR-15/16 can lead to Bcl-2 overexpression, enhancing cell survival and growth [13,14,15]. This pathological development raises the intriguing possibility of counteracting this deleterious effect. Because of this critical role, by synthetically reinstating miR-15/16, there may be a therapeutic potential to thwart the growth and survival of leukemia cells, targeting these pivotal oncogenic pathways [16,17]. However, the challenge in implementing RNA-based therapies is ensuring intracellular delivery, protecting the RNA from degradation, and maintaining tissue specificity for the target gene [18,19]. A potential solution is ferritin-based nanoparticles. These are highly versatile nanotechnological instruments, offering stability in the extracellular environment and efficiency in encapsulating and delivering small nucleic acids to various cancer cell lines [20]. Their rapid internalization through the transferrin receptor (TfR1), often overexpressed in malignant cells, makes them effective for targeted delivery [21,22]. One variant, HumAfFt (humanized ferritin from *Archaeoglobus fulgidus*), offers unique advantages. Modified from ancient ferritin, the HumAfFt structure allows mild and reversible control of assembly–disassembly, which is vital for encapsulating cargo. Its compatibility with human TfR1 facilitates cellular uptake [23,24,25]. Further, customization through chemical modification enables the non-covalent entrapment of miRNA molecules [26]. Previous studies from our group indicate that conjugation with cationic polyamine-based thiol-reactive linkers of the internal cavity of HumAfFt ferritin was crucial and beneficial in the delivery of double-strand siRNA targeting GAPDH expression in various cancer cells [27]. As previously outlined, the cationic linker (PA) was designed as a piperazine-based compound featuring two piperidine rings bearing, at one end, a pentafluoro-benzene-sulfonamide group, which is reactive towards the thiol of cysteines. The PA linker displayed remarkable solubility in aqueous solutions and exhibited the capability to establish strong electrostatic interactions with negatively charged nucleic acids. The siRNA-loaded nanoparticles resulted in the substantial silencing of GAPDH activity, achieving a reduction of over 25%. This outcome bears significant implications for cellular well-being, given GAPDH’s pivotal involvement in energy production, glucose regulation, and various other cellular functions. To underscore the versatility of the ferritin-based system as a nucleic acids nano-vehicle, the PA-HumAfFt was evaluated to transport miR-15/16 into MEG01 cells characterized by the CLL-typical 13q14 deletion [8]. With TfR1 overexpression common in malignant cells, including MEG01, selective targeting towards CLL cells is possible. Our observations confirmed that functionalized ferritin nanoparticles unveil a versatile and multifaceted miRNA carrier that offers numerous advantages over traditional transfection agents, including ease of purification, abundant production, and thermostability. The technology not only safeguards and stabilizes the miRNAs but also facilitates their entry into tumoral cells, enabling precise and targeted delivery of nucleic acids. Notably, the synergistic approach, employing simultaneous administration of multiple miRNAs, led to a significant decrease in cancer cell survival, correlated with reduced Bcl-2 expression. The multifunctionality of cargo encapsulation makes functionalized ferritin nanoparticles a compelling tool for efficient and targeted delivery of nucleic acids in cancer treatment [28,29,30].

## 2. Materials and Methods

### 2.1. Production and Purification of HumAfFt

The sequence of the HumAfFt protein was modified with an M54C mutation per monomer to functionalize the inner cavity with thiol-reactive polyamines. HumAfFt was purified as previously described [24]. Briefly, the humanized ferritin gene is cloned into a pET-22a vector (Novagen) and introduced into BL21-competent cells to allow for the expression of the HumAfFt. Cells are harvested and lysed via a single cycle of sonication for 15 min (3 s on and 6 s off at 60% Amplitude) to release the protein into the soluble fraction. The cell lysate is then centrifuged for 30 min at 18,517× *g*. The clarified lysate is purified through two steps of ammonium sulfate fractionated precipitations (40% and 70%). As a final purification step, the protein is loaded onto a HiLoad 26/600 S-400 column using an AKTA-pure system (Cytiva, Marlborough, UK). The purified protein concentration, as a monomer, was measured via UV spectrum using an extinction coefficient of 32,400 M^−1^cm^−1^ with a Jasco V-750 (Jasco Corp., Tokyo, Japan). Finally, the purified ferritin protein is sterilized via filtration using 0.22 μm filters and kept at 4 °C until use.

### 2.2. Functionalization of HumAfFt with Polyamines-Based Thiol-Reactive Linker

The preparation of PA-conjugated HumAfFt to modify the inner protein cavity with positive charges was previously described [27]. Briefly, a solution of 12 μM HumAfFt in 20 mM HEPES pH 7.5 was reduced using 10 eq. TCEP (Tris(2-carboxyethyl) phosphine hydrochloride) concerning SH group for 1 h at room temperature under mild agitation. Next, 10 equivalents of PA concerning the SH group were added to the protein solution under mild agitation for 16 h at 37 °C (350 RPM) using an Eppendorf ThermoMixer^®^ C (Eppendorf, Milan, Italy). The unreacted compound was removed through PD10 desalting columns (Cytiva, Marlborough, UK), and the final protein concentration was calculated. All cysteine residues were conjugated, leading to the functionalization of all 24 sites modified with PA compound, as demonstrated using QTof Synapt G2-Si mass spectrometry analysis. The PA-HumAfFt complex retained the ability to reverse association in a spherical conformation, as previously reported [27].

### 2.3. Incorporation of miRNAs into PA-HumAfFt

MiR-15a and miR16-1 and a “scramble” sequence used as a negative control, consisting of 22 nucleotides, were synthesized and purified using desalting chromatography by Sigma Aldrich. Two sets of fluorophore-labeled miRNAs were used: 6-FAM-miRNAs and Cy5.5-miRNAs were designed with a fluoresceine and cyanine group, respectively, at the 5′ end of the sense sequence. All miRNAs and fluorophore-labeled-miRNAs were resuspended in sterile 10 mM Tris-HCl pH 7.5 at a final concentration of 100 µM. The functionalized PA-HumAfFt was equilibrated in 20 mM Hepes, pH 7.5, to favor the opening of the protein nanocage and allow for the encapsulation of the negatively charged nucleic acid. MiRNAs were added in slight excess (1.5 eq, 9.7 µM) with respect to the concentration in 24mer, and the solution was kept under constant stirring at 10 °C for about two hours. In the end, 100 mM MgCl_2_ was added to induce the closure of the ferritin in its characteristic spherical shape. Hereafter, the complete sequences are reported. MiR-15a: 5′-UAG CAG CAC AUA AUG GUU UGU G-3′; miR-16-1: 5′-CCA GUA UUA ACU GUG CUG CUG A-3′; negative control (SCR): 5′-GGU UCG UAC GUA CAC UGU UCA-3′.

### 2.4. Native Electrophoresis Gel Analysis and RNase Digestion

To assess the amount of miRNA loaded into the ferritin nanocage, reassembled HumAfFt (360 nM) with 1.5 eq of miRNA (810 nM) were treated with 0.5 ug/uL RNase (Sigma-Aldrich, Milan, Italy) for 30 min at 37 °C to remove free miRNA outside the protein. Then, an RNase inhibitor (Sigma-Aldrich, Milan, Italy) was added to stop the reaction. An identical amount of naked miRNA, naked miRNA after treatment with RNAse, and subsequently with RNAse inhibitor was used as a control. In this experiment, all miRNAs were 6-FAM-labeled at the 5′ end for selective staining and miRNA quantification. All samples were added of 8% (*v*/*v*) glycerol. Electrophoresis was performed under native conditions at a constant voltage of 80 V for 30 min, using a non-denaturing 4% polyacrylamide gel in 0.5× TBE buffer (45 mM Tris-Borate, 1 mM ethylenediaminetetraacetic acid pH 8.6). To stain the RNA, gels were incubated with SYBR Safe (Invitrogen, Waltham, MA, USA) in 30 mL of 0.5× TBE for 10 min, and images were acquired using the Chemidoc MP Imaging System (Bio-Rad, Milan, Italy). The quantification of the signal intensity was carried out using the ImageJ Lab 6.1 software.

### 2.5. Protein Extraction and Western Blotting Analysis

Cells were harvested and lysed with CelLytic (C2978, Sigma-Aldrich), and protein concentration was determined using the Pierce BCA Protein Assay Kit (23225, Thermo Fisher Scientific, Waltham, MA, USA). Twenty µg of proteins were separated through 4–15% gradient SDS- electrophoresis. Briefly, the proteins were then transferred under dry conditions onto nitrocellulose membranes at 25 volts for 7 min and 30 s and incubated overnight at 4 °C with the primary antibody: anti-BCL 2 (dilution 1:800, B3170-2mL; Sigma-Aldrich), and anti-β-actin (dilution 1:5000, sc-47778; Santa Cruz Biotechnology, Dallas, TX, USA). On the next day, they were incubated with the respective horseradish peroxidase-conjugated secondary anti-mouse (1:5000, sc-516102, Santa Cruz Biotechnology) antibody for 1 h at RT. Membranes were washed with PBS- 0.1% Tween 20 and developed using the chemiluminescence system Chemidoc MP Imaging System (Bio-Rad, Milan, Italy).

### 2.6. Cell Cultures and Transfection

Cell lines: MEG-01 cell lines were purchased from ATCC (Manassas, VA, USA). The cells were grown in RPMI-1640 medium (R8758, Sigma-Aldrich), supplemented with 100 IU/mL penicillin-streptomycin (P4458; Sigma-Aldrich), and 10% fetal bovine serum (F7524; Sigma-Aldrich), and kept inside a humidified incubator at 37 °C with 5% CO_2_.

Cell transfection: After trypsinization, MEG-01 cells were plated in a 96-well multiwell (MW96). For each sample, a set of different concentrations (150 nM, 300 nM, 450 nM, and 600 nM) of the miRNA-PA-HumAfFt complex were incubated for 24 and 48 h. The scramble miRNA-loaded ferritin was used as a negative control. The cells were subjected to trypan blue exclusion assay 24 h and 48 h after transfection.

### 2.7. Viability Assay

The growth medium was collected, and the cells were washed once with phosphate-buffered saline (PBS). Adherent cells were removed through treatment with 0.25% trypsin and 2.21 mM EDTA (Euroclone, Milan, Italy), which in turn was blocked using an RPMI-1640 medium (R8758, Sigma-Aldrich, Milan, Italy). All the collected cell fractions were centrifuged for 5 min at 78× *g*, and the supernatant was carefully discarded. The harvested cells were then washed with PBS, centrifuged for 5 min at 78 g, and the supernatant discarded. Following the addition of a 0.4% (*w*/*v*) trypan blue stain solution (EBT-001, NanoEntek), cells were transferred to a cell counting slide (EVS-050, EveTM NanoEntek, Seoul, Republic of Korea), visualized, and counted using the EveTM Automatic Cell Counter, (NanoEntek, Seoul, Republic of Korea), with blue cells considered nonviable. The numbers of viable and non-viable cells were recorded and used to calculate the percentage of cell viability in terms of the ratio between the number of viable cells and the total number of cells.

### 2.8. Cytometric Analysis

For flow cytometry analysis, MEG01 cells were seeded on multi-well plates. Cells were incubated with Cy5.5-labeled miRNA-PA-HumAfFt nanoparticles. MEG01 cells were washed two times with PBS, detached with Trypsin-EDTA (Euroclone, Milan, Italy), washed with Phosphate-Buffered Solution (PBS), and resuspended in BD-FACS Flow buffer. Control cells were treated in the same way but without PA-HumAfFt incubation. Internalization of conjugated nanoparticles before and after TB treatments was measured at the BD LSRFortessa (BD Biosciences, San Jose, CA, USA) equipped with a 639 nm laser and FACSDiva software (BD Biosciences version 6.1.3). Live cells were first gated using forward and side scatter area (FSC-A and SSC-A) plots, then detected in the red channel for Cy5.5 expression (666 nm filter) and side scatter parameters. The gate for the final detection was set in the control sample. Data were analyzed using FlowJo9.3.4 software (Tree Star, Ashland, OR, USA). The detached cells were resuspended in BD-FACS Flow buffer containing anti-TFr1 antibody (Anti-Transferrin Receptor Monoclonal Antibody (3B82A1)) (Thermo-Fisher Scientific, Milan, Italy) (1:50) for 1 h. The cells were then centrifuged at 78× *g* for 10 min as described in the viability assay section, and the supernatant was discarded. Subsequently, the cells were resuspended in BD-FACS Flow buffer with a secondary antibody (Rhodamine Red™-X conjugated AffiniPure F(ab’) 2 fragment goat Anti-Mouse IgG (H-L) (Jackson ImmunoResearch Laboratories, Inc., Ely, UK)) at a dilution of 1:5000 for 1 h. Following another centrifugation step, the cells were resuspended in BD-FACS Flow buffer before being analyzed through flow cytometry using the same settings as described earlier.

### 2.9. Immunofluorescence

MEG-01 cells were seeded on MW96 plates and treated with 300 nM Cy5.5-labeled miRNA-PA-HumAfFt complex for 3 h. Then, Hoechst 33342 (Thermofisher, Waltham, MA, USA) was used to stain nuclei following the manufacturer’s instructions. Samples were acquired using Cytation 1 instrument (Agilent Biotek, Santa Clara, CA, USA) with a 10× magnification. The evaluation of the red fluorescence intensity was conducted using GEN5 software, version 3.13 (Agilent Biotek, Santa Clara, CA, USA).

### 2.10. Statistical Analysis

All the experiments were performed at least 3 times. The statistical analysis was conducted using ANOVA, followed by Bonferroni correction. A *p*-value lower than 0.05 was considered to be statistically significant.

## 3. Results

In the present study, we explored the efficiency of the PA-modified nanoparticles to co-deliver and protect miR-15a and miR16-1 from RNase digestion, restore the correct expression levels of Bcl-2, and induce cancer cell death.

### 3.1. miRNA Encapsulation into Ferritin Nanocages

As described in our previous work [27], the polyamine linker design significantly contributed to the electrostatic attraction and incorporation of siRNA molecules into the HumAfFt at a physiological pH. The chemical bioconjugation was confirmed using mass spectrometry analysis. To evaluate the effectiveness of PA-HumAfFt nanoparticles in both encapsulating miRNAs and protecting them from RNase degradation, we used native electrophoresis, as shown in Figure 1. We trapped the miRNAs—specifically miR-15a and miR-16-1—within the inner cavity of the protein complex. This trapping was facilitated by inducing protein polymerization with magnesium ions (Mg^2+^). After forming the complex, we treated it with RNase at 37 °C for 30 min to remove any unencapsulated miRNAs. An RNase inhibitor was then added to stop further degradation. The loading capacity and the entrapment efficiency of miR within the nanoparticles were measured by following the absorbance of Cy5.5-linked miRNA at 647 nm (ε = 250,000 M^−1^ cm^−1^) and calculated as the ratio between the concentration of miRNA entrapped over the concentration of nanoparticles and the total amount of miRNA used. Our studies revealed a 46.7% loading capacity and 30% entrapment efficiency. As a further confirmation, an electrophoretic mobility shift assay was performed to analyze protein/RNA interaction in extreme experimental conditions in the presence of RNase. As illustrated in Figure 1, where only FITC positive bands are visible, miRNA encapsulated in the internal cavity following the restoring of the HumAfFt closed conformation upon the addition of MgCl_2_ (Lane 1), displays a band shift compared to miRNA with HumAfFt in the open conformation (Lane 2, in the absence of MgCl_2_). After treatment with RNase at 37 °C, the miRNAs that have been effectively incorporated into the ferritin structure are partially protected from RNase degradation, as evidenced in Lanes 3 and 4. In comparison with the initial samples, where the miRNA was loaded in a 1.5-fold excess with respect to the protein concentration, the band intensity is lower due to the expected degradation of the free miRNA unloaded into the cavity. The quantification of the band intensity revealed a reduction of 72% ± 4.8 with respect to the initial samples (Lane 1 and 2). This value is in complete agreement with the analysis performed using the UV–Vis spectrophotometer that confirmed a 30% miRNA encapsulation efficiency. On the contrary, naked miRNa after treatment at 37 °C is subject to complete degradation when exposed to Rnase under experimental conditions (Lane 6) compared to an equal amount of miRNA used as a control (Lane 5). The amount of encapsulated and thus protected miRNA was quantitatively assessed using densitometric analysis, performed with the ImageJ software, and compared to the control RNA (shown in Lane 5). As expected, HumAfFt is not positive for the FITC fluorescence staining (Lane 8 and 9), thus ensuring a selective evaluation of the miRNA present in the samples.

### 3.2. Ferritin Receptor Expression in MEG01 Cell Line

Since ferritin is imported into cells via the transferrin receptor TFR1, we decided to validate the expression of this receptor in our model cell lines that carry the deletion for the miRNAs of interest through cytometric analysis. To accurately establish the expression levels of the TFR1 receptor, we incubated the cells solely with a rhodamine secondary antibody (Jackson ImmunoResearch Laboratories, UK). The dark grey population, which is partially fluorescent in the presence of only the secondary antibody, represents the baseline fluorescence level. The black population is the result of labeling with the primary antibody against the TFR1 receptor followed by the secondary fluorescent antibody. The data indicate that 90% of the analyzed cells exhibit high levels of the receptor in question, confirming that MEG01 cells are an excellent model for the study of miRNA-loaded ferritin particles (Figure 2).

### 3.3. Cellular Uptake of Cy5.5 Labeled miRNA-PA-HumAfFt Nanoparticles

To evaluate the levels of cellular uptake of miRNA-PA-HumAfFt qualitatively and quantitatively, we used fluorescence microscopy and flow cytometry, respectively. The Cy5.5 labeled-miRNA-PAs-HumAfFt system was incubated in MEG01 cells at concentrations of 300 nM and 450 nM for 3 h. The concentrations utilized for these analyses were determined through an initial screening of the biological effect within the range of 150 nM to 900 nM in miRNA concentration. The minimum concentration required to achieve 100% cell death was identified as 300 nM. As a result, the experiments were confined to 300 nM and an elevated concentration (450 nM) to account for potential dependence in encapsulation efficiency. As shown in Figure 3, the Cy5.5 labeled miRNA entrapped into the PA-modified protein successfully enters the treated MEG01 cells as it is localized in the cytoplasm in the perinuclear region, in complete agreement with targeted delivery of HumAfFt [24].

These findings correlated well with the observations made through fluorescence measurement using a flow cytometer for quantitative analysis (Figure 4). FACS analysis confirms a 97.5% cell-positive population for the Cy5.5-labeled miRNA using both concentrations of HumAfFt and demonstrates an almost complete internalization under these experimental conditions.

### 3.4. MiRNA Delivery and Effect on Apoptosis

The delivery of the miRNAs (miR15-a and miR16-1)-loaded HumAfFt nanoparticles was also validated by analyzing the apoptosis induction in MEG01 cells. The extent of cell death after 24 h of treatment with 300 nM miRNA-PA-HumAfFt complex was quantified using the Trypan blue exclusion assay and the percentage of dead cells (Figure 5A). As a negative control, a non-targeting scramble RNA sequence (SCR) was entrapped into PA-HumAfFt. In comparison with the negative control of PA-HumAfFt loaded with a scramble miRNA sequence, cells incubated with miR15-a-PA-HumAfFt still exhibited approximately 70% vitality. Similarly, the PA-HumAfFt complex containing miR16-1 still maintained about 80% vitality at the same concentration. The most surprising effect is obtained using a 1:1 mixture of PA-HumAfFt containing miR15-a and miR16-1. In this case, 100% of the cells died at a concentration of 300 nM. These data were also analyzed as a growth curve over 48 h of incubation at 300 nM miRNA concentration (Figure 5B), displaying a similar behavior for the miR15-a-loaded and miR16-1-loaded nanoparticles (orange and gray curve, respectively) with a decreasing of cell vitality within 24 h. Remarkably, the effect of the co-delivered mixture (150 nM miR15-a and 150 nM miR16-1) showed 100% cell mortality with a rapidly decreasing curve (yellow curve) of live cells within 24 h.

### 3.5. BCL-2 Expression Is Down-Regulated via miR-Pa-HumAfFt Treatment

To validate that the observed cell death was specifically triggered via the delivery of miRNA-PA-HumAfFt, we analyzed the expression of known direct targets of miR-15/16 using Western blot assays. We treated MEG01 cells for 24 h either with miR-15 and miR-16 or with an equivalent quantity of encapsulated scrambled RNA sequences for control. Specifically, we focused on the expression levels of Bcl-2 and Cyclin E, two well-documented targets of miR-15/16 that are closely associated with tumorigenesis [15,31]. Our densitometric analysis, presented in Figure 6, shows significant changes in the expression levels of the target genes. We noted a decrease in Bcl-2 expression ranging from 30% to 60% when using individual miRNAs or the miRNA mix, respectively. Additionally, Cyclin E expression was reduced by approximately 50% with both single miRNAs and the miRNA mix, in comparison to the scramble samples.

## 4. Discussion

In Chronic Lymphocytic Leukemia (CLL), the deletion of the 13q14 chromosome segment is the most prevalent chromosomal aberration, affecting 50–60% of patients [7]. The importance of this deletion is further highlighted by numerous studies focusing on the microRNA miR-15a and miR-16-1 cluster, which is located within the 13q14 region. These microRNAs primarily target the Bcl-2 oncogene, a gene that plays a crucial role in preventing cell death. In the case of CLL, the loss of these microRNAs leads to an overexpression of Bcl-2 in B cells, inhibiting their natural death and consequently causing the accumulation of these cells into the blood of the patients [32]. Currently, the treatment strategy for CLL involves several therapeutic options, including the use of Bcl-2 inhibitors such as venetoclax, addressing the overaccumulation of the Bcl-2 protein [33]. However, this approach has its limitations [34,35]. The altered microRNA cluster affects not just Bcl-2 but also other significant genes such as cyclin D1, cyclin E, MCL1, and WNT3 [36,37,38]. Therefore, treating CLL patients solely with Bcl-2 inhibitors might not fully address the disease complexity, often requiring the use of combined treatment with inhibitors of Bruton tyrosine kinase (BTK) such as ibrutinib and acalabrutinib [39,40,41,42,43,44]. Recent progress in understanding that this pathology is frequently initiated by the loss or addition of chromosomal material, coupled with subsequent mutations, has highlighted the significance of an alternative and potentially more encompassing treatment approach. Our strategy involves reintroducing microRNAs 15a and 16-1, aiming to target a broader spectrum of affected genes and restore the functionality of various cellular mechanisms disrupted by the 13q14 deletion. By doing so, a more effective means of counteracting the multifaceted genetic alterations may lead to the arrest of the increased aggressiveness of leukemia at its onset. In this research, the delivery of the miR15-A and miR16-1 was achieved using functionalized ferritin nanocaged systems. This finding confirmed that humanized ferritin nanoparticles derived from Archaeoglobus fulgidus (HumAfFt) and modified through simple chemical crosslinking with polyamine-based compounds (PA-HumAfFt) are highly effective and functional for small RNA delivery. Considering the RNA incorporation rate into each nanoparticle, which is roughly estimated at 50% loaded molecules, it can be deduced that individual cells are exposed to a restricted quantity of exogenous miRNA. Despite this limitation, loading miRNA into PA-HumAfFt nanoparticles significantly protected it from RNase activity, even in extreme conditions, as demonstrated through electrophoresis analysis where an intense band is still detectable after 30 min treatment with RNase. PA-HumAfFt-based nanoparticles exhibit a remarkable capability for targeting cells originating from leukemia, including MEG01 cells, through the highly expressed TfR1 receptor. As indicated by the fluorescence microscopy experiments, our results strongly highlight a cellular distribution in the cytoplasm of Cy5.5-labeled miRNA-PA-HumAfFt, thus confirming the typical uptake mediated via the TfR1 receptor, showing nearly 100% incorporation efficiency. The delivery of intact miRNA is assessed through the examination of its functional activity, followed by the release of encapsulated miRNA, which significantly impacts cellular viability when either of the two genes is individually restored. As depicted in the growth curve in Figure 5B, MEG01 cells cease growth after a 24-h treatment with 300 nM of delivered miRNA. Notably, in dual treatments, where miR15-a and miR16-1 are co-delivered, an even more pronounced synergistic effect occurs, ultimately resulting in the mortality of the entire population of cancer cells within 24 h. These results are compelling evidence that the induced cell death was indeed mediated by the targeted delivery of miRNA-PA-HumAfFt through the TfR1 receptor, as demonstrated using fluorescence microscopy. Additionally, the functionality of miRNA was demonstrated by restoring the natural propensity of apoptosis, as evidenced by the decrease of 60% in Bcl-2 expression levels. This also has a great impact on the expression of Cyclin E, as demonstrated through a 50% reduction in expression after treatment with 300 nM miRNA-loaded nanoparticles, providing a prominent example of the multifunctionalities and versatility of such ferritin-based systems targeting miR15-a and miR16-1 genes.

## 5. Conclusions

Ferritin-based nanocages possess unique attributes that can be fine-tuned through straightforward chemical crosslinking, tailoring their properties to accommodate specific cargo molecules. In this study, modified PA-HumAfFt nanoparticles were successfully prepared under physiological conditions with efficient entrapment of miRNAs targeting oncogenic pathways. In agreement with previous studies [27,30], our findings confirmed that PA-modified ferritins are a remarkable nanotechnological platform that is well suited for a wide range of actions for the safe delivery of small nucleic acids through the TfR1 receptor. In this scenario, where the correlation between the existence of TfR1 and cancer cells exhibiting a genetic material deletion is established, the system demonstrated remarkable effectiveness in restoring physiological conditions. This highlights the intricate interplay and the potential of miRNA to influence cell behavior, even in small quantities. In conclusion, our outcome led to an enhanced and efficient impact, thereby reinforcing the notion that the efficacy of miRNA transfer depends on a diverse range of cellular targets, some of which are likely yet to remain unidentified.

## Figures and Tables

**Figure 1 pharmaceutics-16-00402-f001:**
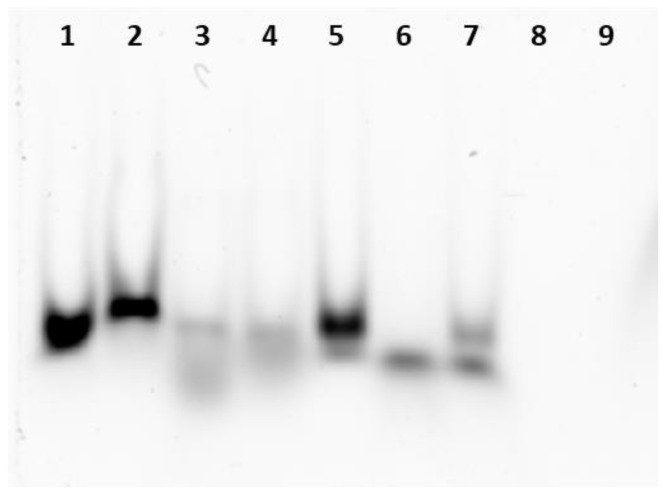
Native Electrophoresis on polyacrylamide gel: (1) open HumAfFt + miRNA, (2) closed HumAfFt + miRNA, (3,4) HumAfFt/MgCl_2_ + miRNA+ 0.5 ug/uL RNAse, (5) naked miRNA, (6) miRNA + RNAse, (7) miRNA + RNAse + Rnase inhibitor, (8) open HumAfFt, and (9) closed HumAfFt.

**Figure 2 pharmaceutics-16-00402-f002:**
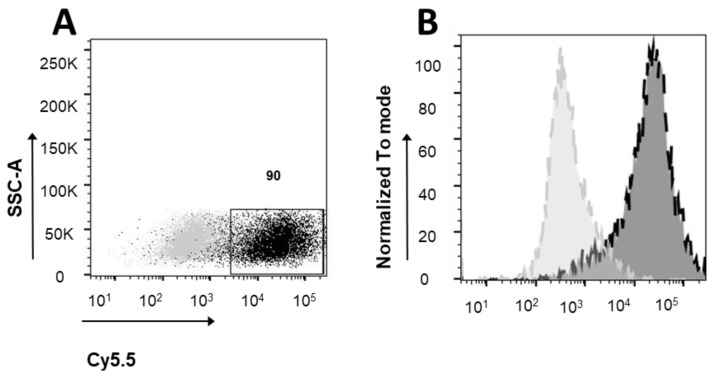
Flow cytometry analysis of TFR-1 expression level in MEG-01 cells. Figures show the overlay of dot plots (**A**) or histograms (**B**) of CTRL secondary Ab (light grey) and TFR-1 expression level (black, Cy5.5 positive cells). The gate indicates the percentage of TFR-1-positive cells (90). For each sample 10,000 events gated on live cells were acquired.

**Figure 3 pharmaceutics-16-00402-f003:**
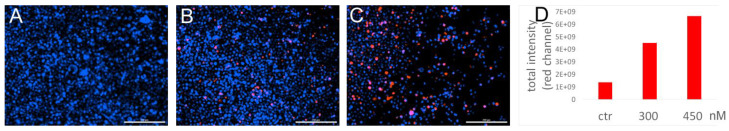
Fluorescence microscopy of live MEG01 cells control (**A**), and after treatment with 300 nM (**B**) or 450 nM (**C**) of miRNA-PA-HumAfFt (red) after 3 h of incubation at 37 °C. Hoechst 33342 (blue) was utilized to stain the nuclei of the cells (scale bar = 200 μm). In (**D**), the evaluation of the red fluorescence intensity in the images was conducted using GEN5 software, version 3.13.

**Figure 4 pharmaceutics-16-00402-f004:**
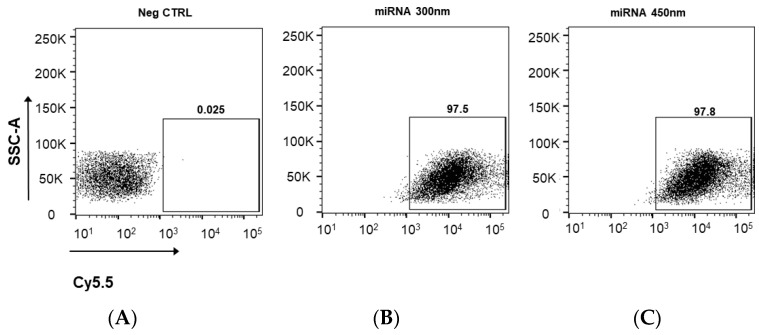
Flow Cytometry analysis of the negative control (**A**), 300 nM (**B**), and 450 nM (**C**) Cy5.5-miRNA-PA-HumAfFt uptake after 3 h of incubation. Gates indicate the percentage of Cy5.5 positive cells. For each sample, 30,000 events gated on live cells were acquired.

**Figure 5 pharmaceutics-16-00402-f005:**
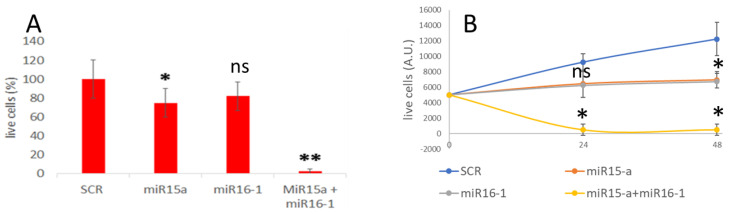
Cell viability after miRNA-PA-HumAfFt treatment. (**A**) Histogram plots depict the percentage of live cells after 24 h of incubation of untreated cells (CT) and those treated with scramble miRNA sequence loaded into PA-HumAfFt, miR-15a loaded-HumAfFt, miR-16-1 loaded-PA-HumAfFt, and 1:1 miR15a- and miR16-loaded nanoparticles at 300 nM miRNA concentration. (**B**) The growth curve of MEG01 cells is shown at 300 nM miRNA concentration. The graph presents the cell count following 24 and 48 h of the specified treatments. The cell number was determined through the trypan blue exclusion assay. * = *p* < 0.05; ** = *p* < 0.001; ns = non-significant.

**Figure 6 pharmaceutics-16-00402-f006:**
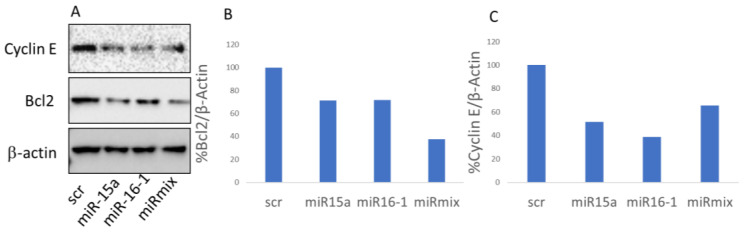
(**A**) Western blot analysis for the measurement of Bcl2 and Cyclin E expression levels with a scramble sequence (scr), miR15a, miR16-1, or with a 1:1 mixture of miR15a and miR16-1-loaded PA-HumAfFt (mirmix). β-actin was used as a loading control. Densitometric quantification of the western blot in A for Bcl2 (**B**) and Cyclin E (**C**).

## Data Availability

Data are contained within the article.

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
