# Peer review of "Combined Delivery of miR-15/16 through Humanized Ferritin Nanocages for the Treatment of Chronic Lymphocytic Leukemia"

_pharmaceutics, 2024, doi:10.3390/pharmaceutics16030402_

Round 1

Reviewer 1 Report (Previous Reviewer 2)

Comments and Suggestions for Authors

The manuscript has been revised. I have no further comments.

Reviewer 2 Report (Previous Reviewer 3)

Comments and Suggestions for Authors

Authors approved all my suggestions, thank you. So I believe that the manuscript is acceptable for publication

Comments on the Quality of English Language

NA

This manuscript is a resubmission of an earlier submission. The following is a list of the peer review reports and author responses from that submission.

Round 1

Reviewer 1 Report

Comments and Suggestions for Authors

The work is interesting in preparing nanocage from ferrritin for miRNA and examinations of its effect on special cells line. It would be interesting to try it on CLL cells. How to direct these nanocages on CLL leukemic cells in human body? Nevertheless work is interesting that shows possibilities of treatment CLL that's why I think it can be printed without corrections. 

Reviewer 2 Report

Comments and Suggestions for Authors

The present article by Liberati et al titled " Combined delivery of miR-15/16 through Humanized ferritin nanocages for the treatment of chronic lymphocytic leukemia" is well written and scientifically important.

I have following comments:

Figure 1: Why is band intensity decreased so much after treating the closed HumAfFt + miRNA complex with RNase. Which is actually supposed to degrade only the unencapsulated miRNA. I also assume that the migration pattern for lane 3 and 4 should resemble lane 2. 

Will it be possible for the authors to add a few reference proteins with known migration patterns?

How many technical and biological replicates were done for the experiments?

Reviewer 3 Report

Comments and Suggestions for Authors

The authors present an interesting paper regarding the use of targeting mir15-16 via nanoparticles loaded with siRNA.

The work is interesting, considering the possibility of therapeutic implications linked to the targeting of mir15-16 and the effect on apoptosis and viability of leukemic tumor cells.

Aside from minor spelling checks, here are some comments:

Minor points: 

line 94 “it is expressed and purified as previously described”. If the subject is the HumAfFt protein, please rewrite the sentence concisely. 

Line 97 specify pulse, amplitude, time, and number of cycles to allow reproducibility info

Line 98 avoid using RPM as RPM values are radius-dependent. Translate in G (same thing in lines 170 and 171)

Line 99 specify in what the ammonium sulfate fractionated precipitations consist.

Line 102 “The concentration of the purified protein in its monomer status was measured by UV spectrum using a Jasco V-750 (Jasco Corp, Tokyo- Japan)”

103 use the “micro” symbol instead of “u” – has the purity evaluated together with the concentration? If yes with which technique?

109 what does “mild” agitation means? Indicate agitator model and RPM used for consistency

Line 112, substitute Cytiva with Cytiva, MA – USA

Line 137 specify the FITC labeling. 

149 include transferring details (voltage, dry or wet?) 

151, substitute beta with the correspondent symbol and add “anti” 

Line 162 The authors should include concentrations after “different concentration” and not after the timepoint indication.

Line 164 cells were subjected to trypan blue exclusion assay.

Line 169 which complete medium was used?

168 add trypsin concentration and brand (should be Euroclone according to line 182?)

173 When citing a company, always indicate the headquarters localization (NanoEntek, Seul KR) 

175 all the cells are stained with trypan blue, but only death cells are blue. Modify accordingly (i.e. … and blue cells were considered nonviable) 

182 add % of trypsin. 

271 substitute nm with nM (450 nM) 

LINE 244 – cells were incubated with a secondary antibody. Which secondary antibody?

A and B should be indicated in the figure. I don’t particularly appreciate showing the secondary antibody data, which is useless. Moreover, if 90% of cells are positive for TFR1 where is the 10% not expressing this receptor and therefore negative for the fluorescence signal at the flow? 

Figure 5 – The authors should indicate ns where is non-significant and the p-value where is significant. 

Line 308, decide if use Meg 01 or MEG01

Line 311 Add reference referring to “well-documented targets”. 

Figure 6 –

Please add a histogram reporting quantification results to show the results of the densitometric analysis mentioned in line 312.. There also is the space on the right. Then, remove the Fold Change numbers referring to the analysis for a better visualization.  Moreover, under the WB bands, the FC is reported, whether in the text the authors refer to a %. Please modify accordingly for consistency. 

Major points: 

TFR1 expression: add Western Blot results. 

Figure 2: 

A and B should be indicated in the figure. I don’t particularly appreciate showing the secondary antibody data, which is useless. Moreover, if 90% of cells are positive for TFR1 where is the 10% not expressing this receptor and therefore negative for the fluorescence signal at the flow? Using that type of representation this data is missing.

Figure 3: 

Why in Figure 3B there are more cells compared to the control and Figure 3C? Was the same number of cells used for the staining? If the highest concentration has an anti-proliferative effect, how can be possible to have an equal number of blue-positive cells in Figure 3C compared to Figure 3A?  moreover, since the authors stated that in Figure 3B 300nM and in Figure 3C 450nM were used, why the signal intensity for the red color looks the same? Please add histograms showing signal quantification. 

In the methods section was indicated that cells were treated with 300nM, please add 450nM since Figure 3 reports a treatment with 450nM. 

Figure 4: based on what the authors chose the concentration to be used? Is there a better tittering available? The difference between 300nM and 450nM (0.3%) looks weird compared to the 150nM. 

Figure 6 if the authors showed the impact of single miRNA delivery on cell growth, also the effect on BCL2 and Cyclin E expression for the single miRNA should be shown together with the effect of the co-delivery. 

Why don’t add also Cyclin D1, MCL1, and WNT3 since they are cited in the discussion together with Cyclin E and BCL2?

Even the argument and the aim of the work is well conceived, it is very poorly presented. I believe that authors must improve paper with more convinced data before re-submission 

Comments on the Quality of English Language

minor spelling checks